# Facile Synthesis of Tricyclic 1,2,4-Oxadiazolines-Fused Tetrahydro-Isoquinolines from Oxime Chlorides with 3,4-Dihydroisoquinoline Imines

**DOI:** 10.3390/molecules27103064

**Published:** 2022-05-10

**Authors:** Kaikai Wang, Yanli Li, Wei Zhang, Rongxiang Chen, Xueji Ma, Mingyue Wang, Nan Zhou

**Affiliations:** 1School of Pharmacy, Xinxiang University, Xinxiang 453000, China; wangkaikai@xxu.edu.cn (K.W.); xinxiangzhangwei@126.com (W.Z.); wmyoutstanding@163.com (M.W.); zhounan5180708@163.com (N.Z.); 2Key Laboratory of Nano-Carbon Modified Film Technology Engineering of Henan Province, Xinxiang 453000, China; 3Medical College, Xinxiang University, Xinxiang 453000, China; liyanli2005@126.com

**Keywords:** 1,2,4-oxadiazolines, oxime chlorides, cycloaddition, cyclic imines

## Abstract

A mild and efficient strategy for the synthesis of tricyclic 1,2,4-oxadiazolines-fused tetrahydro-isoquinolines derivatives via [3 + 2] cycloaddition reaction is reported. The reactions provided the functionalized tricyclic 1,2,4-oxadiazolines in high yields (up to 96%). This protocol is simple and easy to handle. Moreover, a gram-scale experiment further highlights the synthetic utility. The chemical structure of the product was determined by X-ray single-crystal structure analysis. A possible mechanism for this transformation is proposed to explain the reaction process.

## 1. Introduction

The 1,2,4-oxadiazole ring is an important structural motif, but also considered a privileged building block in a variety of medicinal molecules and biologically active compounds [1,2,3,4]. Notably, various functionalized 1,2,4-oxadiazole derivatives possess a wide range of pharmacological and biological activities, such as anticancer, antimicrobial, antiviral, anti-Alzheimer’s disease, and antibacterial activities. [5,6,7,8]. Some representative bioactive compounds containing a 1,2,4-oxadiazole scaffolds are shown in Figure 1.

Due to their importance, a variety of strategies have been developed for the synthesis of 1,2,4-oxadiazolines [9,10,11,12,13,14]. These methods include a general route for the synthesis 1,2,4-oxadiazolines from nitrile oxides with an imine via 1,3-dipolar cycloaddition reaction [9,15]. For example, Lin and co-workers describes a wide range of tricycle 1,2,4-oxadiazole[4,5-*a*]indolone derivatives. These compounds are produced in good yields by the reaction of indolin-2,3-dione in lactime form with phenyl nitrile oxide in the presence of different bases [9]. The 1,3-dipolar cycloaddition reactions have proved to be one of the most effective and well-established methods for the single-step construction of five-membered heterocycles [16,17]. However, the imines compounds, especially for alkyl imines, are rather labile substrates for their preparation [15]. Another approach to the construction of 1,2,4-oxadiazole skeletons is produced by the condensation of amidoximes with a carbonyl compound [18,19,20,21]. These condensation reactions existed under often harsh reaction conditions. Therefore, the development of a mild and efficient synthetic method for the construction of diverse functionalized 1,2,4-oxadiazole skeletons continues to be important and highly desirable in the organic synthetic community.

Tetrahydroisoquinoline units play a pivotal role in natural alkaloid and have found widespread application in antitumor agents [22,23,24]. The 3,4-dihydroisoquinolines are not only used as synthetic building blocks and encompass a great number of biological activities, but, as stabilized cyclic imines compounds, have also been broadly used as versatile synthons in organic synthesis [25,26]. This synthon could react with a variety of nucleophilic reagent to form functionalized potential biological active tetrahydroisoquinoline derivatives [27,28,29,30,31]. Therefore, based on the utility of the isoquinoline units and the 1,2,4-oxadiazolines framework, we hypothesized that a dipolar [3 + 2] cyclization could deliver tricyclic 1,2,4-oxadiazolines-fused tetrahydro-isoquinolines derivatives from nitrile oxide with stabilized cyclic imines. It is worth noting that the tricyclic 1,2,4-oxadiazolines fused tetrahydroisoquinolines derivatives were synthesized by Cho’s group through organocatalytic oxidative cyclization of amidoximes in 32% yield under 40 °C temperature for 12 h (Figure 1a) [32]. Moreover, the amidoxime substrates required an additional synthetic step for their preparation. Thus, efforts to modernize the synthetic methods are necessary. Aiming to develop potent drugs with a range of biological activities, we incorporated the tetrahydroisoquinoline moiety into pharmaceutically privileged structural motifs, for example 1,2,4-oxadiazole skeletons. Herein, we describe that the oxime chlorides reacted with cyclic imines under mild reaction conditions via [3 + 2] cycloaddition reaction, providin convenient and efficient access to potentially bioactive tricyclic 1,2,4-oxadiazolines-fused tetrahydroisoquinolines derivatives (Figure 1b). Moreover, most of the substituted 3,4-dihydroisoquinolines are readily accessible.

## 2. Results

To optimizethe reaction conditions, we initially attempted to react 3,4-dihydroisoquinoline **2a** with in-situ-generated nitrile oxide via dehydrochlorination of the phenylhydroximoyl chloride **1a**. The phenylhydroximoyl chloride **1a** has extensive utility in 1,3-dipolar cycloadditions for the synthesis of a wide variety of important heterocycle compounds, which could generate nitrile oxides in situ in the presence of base [33,34,35,36,37,38,39]. Gratifyingly, the tricyclic 1,2,4-oxadiazolines-fused tetrahydro-isoquinoline **3a** was formed in 72% isolated yield in the presence of 1,4-diazabicyclo[2.2.2]octane (DABCO) in CH_2_Cl_2_ at room temperature for 12 h (entry 1, in Table 1) via [3 + 2] 1,3-dipolar cycloaddition reaction. Then, the various types of bases were screened to further improve the product yield (entries 2–6). The Cs_2_CO_3_ base showed best results in this [3 + 2] cycloaddition reaction (95% yield, entry 5). The other base, including TEA, DBU, and an inorganic base including Na_2_CO_3_, NaOH, gave the desired product **3a** in 75%, 81%, 82% and 92% yields, respectively. Subsequently, a series of different solvents were tested (entries 7–15). The changing of solvents was ineffective and did not further increase the reaction yield. For instance, the reaction could offer product **3a** in good yield in CHCl_3_, DCE, CH_3_CN, or EtOAc (entries 7–8, and 10–11). Moderate yields of the product were obtained when the reaction was performed in toluene, acetone, Et_2_O, THF or dioxane (entries 9, 12–15). Moreover, the yield was obviously affected when the reaction time was further reduced (entry 16).

## 3. Discussion

With the established optimal reaction conditions, the scope of this [3 + 2] cycloaddition reaction between oxime chlorides and cyclic imines was tested under optimal conditions. The results are summarized in Figure 2. Initially, 3,4-dihydroisoquinoline **2a** was fixed as a substrats to investigate a variety of substituted oxime chlorides **1** for the current reaction. The [3 + 2] cycloaddition reaction could process smoothly and was well tolerated by the various tested oxime chlorides **1** with different electron properties and substitution patterns. The expected cycloadducts were isolated in excellent yields (90–96%), regardless of the positions or electron-donating or electron-withdrawing functional groups of the substituents on the phenyl ring of the R moiety (**3a**–**3k**) (see Appendix A). The results showed that the steric hindrance or electronic nature of R group hardly effected the transformation. On the other hand, when the R groups were heterocycle, the cycloaddition reaction could proceed smoothly, without obvious interference, to provide corresponding cycloadduct **3l**–**3n** in 90%, 91% and 90% yields, respectively. Notably, the chemical structure of **3j** (CCDC 2160406) [40] was unequivocally confirmed by X-ray crystallographic analysis (see Appendix A). Subsequently, a variety of 3,4-dihydroisoquinoline, with different substituents at the C5−C8 positions of the phenyl ring, were examined under our standard, offering the corresponding products **3o**–**3s** in excellent yields (88−95%). There was no obvious effect on the expected products when using 3,4-dihydroisoquinoline derivative as the substrate.

It is worth noting that the heterocyclic imine was compatible in this reaction, providing the desired product **3t** in 93% yield, which demonstrates that the reaction is not limited to aromatic heterocyclic imine. α-substituent on the imine **2u**, with steric hindrance, also reacted well with **1a** in this [3 + 2] 1.3-DCs reaction, offering the desired tricyclic 1,2,4-oxadiazolines-fused tetrahydro-isoquinolines **3u** containing a quaternary stereocenter in excellent yield (96%).

The synthetic utility of the protocol was further highlighted by conducting a gram-scale experiment for tricyclic 1,2,4-oxadiazolines under the standard conditions. The [3 + 2] cycloaddition reaction was performed using 6 mmol of phenylhydroximoyl chloride **1a** and 5 mmol 3,4-dihydroisoquinoline **2a**, affording the corresponding product **3a** in 92% yield without an obvious loss of efficiency (Figure 2).

On the other hand, the stable nitrile oxide **1a’** could be isolated from the corresponding oxime chloride [41,42,43]. The reaction can smoothly take place when the stable nitrile oxide **1a’** reacts with cyclic imines **2** under the standard condition, furnishing the expected product **3v** and **3w** in 93% and 95% yields, respectively (Figure 3).

Based on the experimental results and previous reports [9,15], a possible mechanism for this transformation is proposed to explain the reaction process, as depicted in Figure 4. Initially, the highly active nitrile oxide **A** was formed in situ in the presence of base via dehydrochlorination from oxime halides **1**. Then, this active intermediate **A** could react with the cyclic imines **2** to obtain the desired product **3** via [3 + 2] cycloaddition reaction.

## 4. Materials and Methods

NMR data were obtained for ^1^H at 400 MHz MHz, and for ^13^C at 100 MHz. Chemical shifts were reported in ppm from tetramethylsilane with the solvent resonance as the internal standard in CDCl_3_ solution. ESI HRMS was recorded on a Waters SYNAPT G2. Column chromatography (Waters Corporation, Milford, MA, USA) was performed on silica gel (200–300 mesh) eluting with ethyl acetate/petroleum ether. TLC was performed on glass-backed silica plates. UV light, I_2_, and solution of potassium permanganate were used to visualize products. Petroleum ether and ethyl acetate were distilled. THF was freshly distilled from sodium/benzophenone. Unless otherwise noted, experiments involving moisture- and/or air-sensitive components were performed under a positive pressure of argon in oven-dried glassware equipped with a rubber septum inlet. Dried solvents and liquid reagents were transferred by oven-dried syringes.

The hydroximoyl chloride **1** [44,45,46] and isolable 2,4,6-trimethylbenzonitrile oxide **1a’** [41,42,43] were prepared according to the literature procedures. Cs_2_CO_3_ (0.22 mmol) was added to a solution of oxime chlorides **1** (0.22 mmol), cyclic imines **2** [47,48,49] (0.2 mmol) in CH_2_Cl_2_ (1 mL). The solution was stirred at rt for 12 h. After completion, product **3** was obtained by flash chromatography on silica gel (petroleum ether/ethyl acetate = 15:1 to 10:1).

## 5. Conclusions

In conclusion, we developed a mild and efficient method of preparing a broad range of functionalized tricyclic compounds combining the tetrahydroisoquinoline motif with 1,2,4-oxadiazolines scaffolds in high yields (up to 95% yield) from oxime chlorides with cyclic imines. Additionally, the gram-scale of synthesis of tricyclic 1,2,4-oxadiazoline could further highlight our method’s utility. The described methodology is available, including the starting materials, mild reaction conditions, reaction tolerance for broad functional groups, and convenient operation. The further application of this method is presently under bioactive investigation in our laboratory.

## Data Availability

Data is contained within the article or Appendix A.

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
