# Peer review of "Facile Synthesis of Tricyclic 1,2,4-Oxadiazolines-Fused Tetrahydro-Isoquinolines from Oxime Chlorides with 3,4-Dihydroisoquinoline Imines"

_molecules, 2022, doi:10.3390/molecules27103064_

Round 1

Reviewer 1 Report

Facile Synthesis of Tricyclic 1,2,4-Oxadiazolines-Fused Tetrahydroisoquinolines from Oxime Chlorides with 3,4-Dihydroisoquinoline imines

The work is very interesting, the formation of 1,2,4-Oxadiazolines-Fused Tetrahydroisoquinolines using Cs2CHO3 using CH2Cl2 at room temperature, with excellent yields, is proposed.

The abstract of the work is weak, it is necessary to tell a little more about the results obtained.

Discussion

The discussion of results is not supported by bibliographic references, you have to document each sentence of the discussion.

I suggest that this article be reviewed again by the authors, the abstract and discussion part.

Conclusions

Not comment

Author Response

We sincerely thank the comments by reviewer. We have revised the manuscript based on the suggestion of reviewer:

1) Comments: The abstract of the work is weak, it is necessary to tell a little more about the results obtained.

Response: Thanks for the suggestion! We revised the abstract.

2) Comments: The discussion of results is not supported by bibliographic references, you have to document each sentence of the discussion.

 I suggest that this article be reviewed again by the authors, the abstract and discussion part.

Response: Thanks for the suggestion! We added the bibliographic references based on helpful comments from the reviewer.

Reviewer 2 Report

The authors report the facile synthesis of tricyclic 1,2,4-oxadiazolines-fused tetra-2-hydro-isoquinolines from oxime chlorides with 3,4-dihydroisoquinoline imines. This is a well-written and executed study worthy of publication. I have a few suggestions/corrections for the authors to address:

  • The formula of Cs2CO3 above the arrow in Table 2 is wrong.
  • Delete either “the” or “this” in the sentence  in line 116.
  • “And” should be no bold in line 157.
  • Instead of “is” should be “are” (lines 45, 46, etc.).
  • In my opinion, the phrase “Based on the crystallographic information” sounds non-scientific (line 161).
  • For the compound 3b, there is not enough aromatic protons in 1H NMR spectra. The authors have to check the spectra accurately.

Author Response

Comments: 
The authors report the facile synthesis of tricyclic 1,2,4-oxadiazolines-fused tetra-2-hydro-isoquinolines from oxime chlorides with 3,4-dihydroisoquinoline imines. This is a well-written and executed study worthy of publication. I have a few suggestions/corrections for the authors to address:

(1) We sincerely thank the comments by reviewer. We have revised the manuscript based on the suggestion of reviewer:

1) Comments: The formula of Cs2CO3 above the arrow in Table 2 is wrong.

Response: Thanks for the suggestion! We revised the spelling mistake.

2) Comments: Delete either “the” or “this” in the sentence in line 116.

Response: Thanks for the suggestion! We deleted “the” in the sentence based on helpful comments from the reviewer.

 3) Comments: And” should be no bold in line 157.

Response: Thanks for the comments. We revised this mistake.

4) Comments: Instead of “is” should be “are” (lines 45, 46, etc.).

Response: Thanks for the comments. We revised the sentence based on helpful comments from the reviewer.

5) Comments: For the compound 3b, there is not enough aromatic protons in 1H NMR spectra. The authors have to check the spectra accurately.

Response: Thanks for the comments. We check the spectra accurately and add the correct spectra in the Supporting Information. 

Reviewer 3 Report

                The manuscript proposes an effective process for the synthesis of a wide range of tricyclic 1,2,4-oxadiazolines-fused tetra-2 hydro-isoquinolines with potential uses in medicine.

                The manuscript introduction has an oversized volume with a low relevance related to the material described by the authors. In fact, it almost reaches half of the rest of the manuscript. It is a large volume related to the use of products similar to those reported in the manuscript.  At the same time, a number of references describe [3+2] cycloaddition which produces isoxazole derivatives and not 1,2,4-oxadiazole derivatives.

                The proposed procedure described in the manuscript for generation of tricyclic 1,2,4-oxadiazolines-fused tetra- 2 hydro-isoquinolines is based on [3+2] cycloaddition. The standard compound 3a obtained by authors was previously obtained by organo-catalytic oxidative cyclization of amidoximes and was fully characterized (ref 32). The starting compounds used in the present manuscript are nitrile oxides generated in situ from hydroximoyl chlorides and 3,4-dihydroisoquinolines.

Obtaining of nitril oxides is largely described in the literature and was properly cited by the authors. However, as mentioned above, their [3+2] cycloaddition reaction was used mainly together with C=C bond for the synthesis of individual or fused isoxazole ring. Ref. 15 exemplifies the construction of the 1,2,4-oxadiazole ring using nitril oxides together with imines. Unfortunately, the authors did not detail the content of ref 9 which describes a wide range of tricycle 1,2,4-oxadiazole[4,5‑a]indolone derivatives. These compounds are produced in good yields by reaction of indolin-2,3-dione in lactime form with phenyl nitrile oxide in the presence of different bases. The molecular network of these compounds is relatively closely related to that of the compounds described in the manuscript; therefore, I think the authors need to pay more attention to the content of ref 9. As an observation, ref. 9 reports the formation of nitrile oxide dimers as by-products. Under the conditions used in the manuscript, don’t they appear even in traces?

                In Table 1 the use of Cs2CO3 affords the product 3a in 95 % yield and, using NaOH, the yield “decreases” to 92 %. For a possible implementation of the proposed synthesis hydroxide is incomparably more preferred compared to Cs2CO3. That's why it might have been interesting to use also NaOH in the products syntheses as well to compare the obtained results. The yield variation between 90 % and 96 % seems too small to justify discussions about the influence of different structure factors on the course of the reaction.

                The synthesis procedure is correctly described. However, in order to better characterize the products, it is generally recommended that the 1 H NMR signals be individualized.

Author Response

We sincerely thank the comments by reviewer. We have revised the manuscript based on the suggestion of reviewer:

1) Comments: The manuscript introduction has an oversized volume with a low relevance related to the material described by the authors. In fact, it almost reaches half of the rest of the manuscript. It is a large volume related to the use of products similar to those reported in the manuscript.  At the same time, a number of references describe [3+2] cycloaddition which produces isoxazole derivatives and not 1,2,4-oxadiazole derivatives.

Response: Thanks for the suggestion! We revised the sentence and developed the low relevance sentence based on helpful comments from the reviewer.

2) Comments: Unfortunately, the authors did not detail the content of ref 9 which describes a wide range of tricycle 1,2,4-oxadiazole[4,5 a]indolone derivatives. These compounds are produced in good yields by reaction of indolin-2,3-dione in lactime form with phenyl nitrile oxide in the presence of different bases. The molecular network of these compounds is relatively closely related to that of the compounds described in the manuscript; therefore, I think the authors need to pay more attention to the content of ref 9.

Response: Thanks for the suggestion! We added the relevance sentence based on helpful comments from the reviewer.

 3) Comments: As an observation, ref. 9 reports the formation of nitrile oxide dimers as by-products. Under the conditions used in the manuscript, don’t they appear even in traces?

Response: Thanks for the comments. The substrate of the nitrile oxide could form dimer. However, the reaction could proceed smoothly to observe for products without interference by the forecasted homodimerization. This is comparable with the other method reported in the literature (J. Am. Chem. Soc. 2017, 139, 8661-8666), which the nitrile oxide dimers did not completely interference their reactions.

4) Comments: The synthesis procedure is correctly described. However, in order to better characterize the products, it is generally recommended that the 1 H NMR signals be individualized.

Response: Thanks for the comments. We check the spectra accurately and add the correct spectra in the Supporting Information, such as 3b.

Round 2

Reviewer 1 Report

All changes suggested by the reviewers were made. Highlighting the versatility of the reaction and the excellent yields. I suggest accepting this work in its present form.